# LARGE LANGUAGE MODELS CAN DESIGN GAME-THEORETIC OBJECTIVES FOR MULTI-AGENT PLANNING

## ABSTRACT

Game theory is a powerful paradigm to describe the interplay between participants in interactive multi-agent scenarios, and relies on the knowledge of player objectives or payoff structures for game optimal decision making. However, designing such objectives for games is challenging as it requires evaluating the impact of an agent's actions on the behavior of others, and understanding the effect of changes in one's policy on the behavior of others. Indeed, aligning objective representations with a desired multi-agent behavior is achieved via tedious and impractical heuristics or human trial-and-error. This work aims to ease this process and proposes a multi-agent planning architecture that relies on a large language model (LLM) as the game formulation designer. First, we exhibit the zero-shot proficiency of the more capable LLMs (such as GPT-4) in tuning continuous objective function parameters in accordance with a specified high-level goal for autonomous driving examples. We then develop a planner which uses an LLM as a matrix game designer, for scenarios with discrete and finite action spaces. Given a scene history, the actions available to each agent, and high-level objectives (expressed in natural language), the LLM evaluates the payoffs associated with each combination of actions. From the game structure obtained, agents execute Nash optimal actions, the scene is re-evaluated, and the process is repeated. We evaluate our approach on a heterogeneous robot planning task inspired by wildlife conservation, as well as a household multi-humanoid transport task, and show the superiority of our LLM-based approach to other baselines [1].

## 1 INTRODUCTION

The game-theoretic approach to multi-agent planning delves into the intricate dynamics that arise when multiple autonomous agents interact within a shared environment. In this framework, each agent is assumed to be a rational decision-maker, striving to maximize its own utility or benefit. It provides a formalized and mathematical foundation for predicting and understanding the behavior of multiple agents. By modeling their interactions using game theory, one can analyze the potential outcomes, equilibria, and strategies that agents might employ. The significance of this approach is underscored by its applications in diverse fields like economics (Samuelson, 2016), robotics (LaValle, 2000), transportation (Fisk, 1984), and artificial intelligence (Shoham et al., 2007).

The contemporary landscape of multi-agent systems demands innovative, adaptive, and scalable solutions. The integration of large language models (LLMs) into game-theoretic approaches presents a compelling answer to this request. Here's why:

**Automated Game Design and Harnessing Domain Expertise:** Traditional game-theoretic research often focuses on solving games with predefined structures and objectives, yet the crux of real-world utility lies in designing these very frameworks. Current methods, reliant on iterative human input, often requires meticulous manual efforts and fails to scale. LLMs can revolutionize this, offering automation in design and ensuring that domain-specific expertise is seamlessly incorporated with minimal human efforts, leading to outcomes that resonate more profoundly with real-world scenarios in an efficient and scalable manner.

---

[1]Videos can be found at the anonymous website: https://sites.google.com/view/game-llm

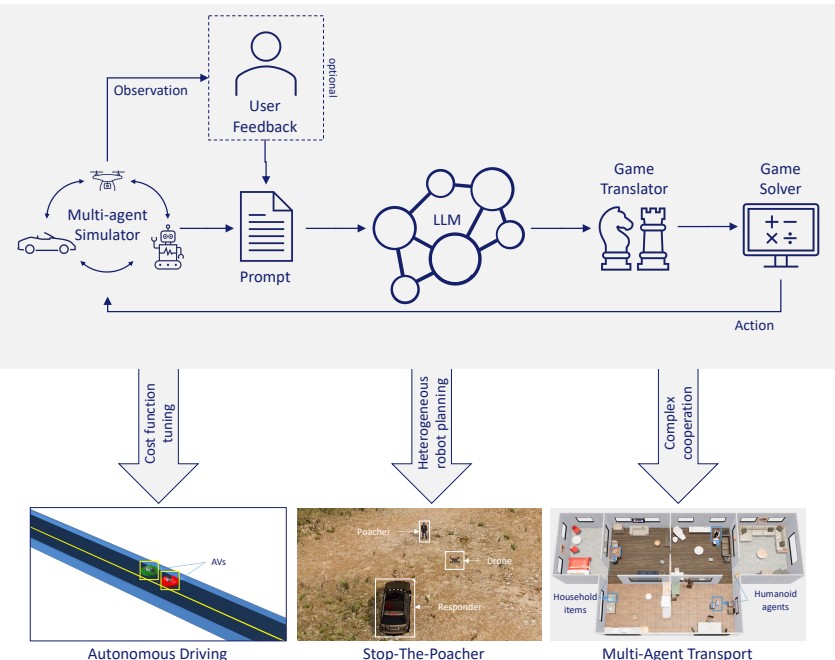

Figure 1: Game LLM: a game-theoretic objective designer. Our framework is amenable to both assisting human tuning as well as designing payoffs for planning in dynamic scenarios.

**Tackling Computational Challenges:** Some games, even with known game structures, by virtue of their complexity, resist optimal solutions through conventional methods and limited computations. In such cases, heuristics or approximations offer a way out, often driven by the nuanced intuition that human bring to the table. LLMs bridge the gap by emulating human-like problem-solving strategies, providing high-level guidelines toward solving complex and intractable games.

**Addressing the Unexpected and Promoting Adaptability:** No model, however sophisticated, can operate flawlessly in every possible scenarios, and unexpected situations are inevitable. LLMs can offer dynamic solutions based on their vast knowledge and common sense reasoning capabilities, and augment human interventions in navigating these uncharted regimes. Furthermore, just as humans evolve strategies based on experiences, LLMs can mimic this adaptability, refining strategies and enhancing the overarching game-theoretic model on the fly through trial-and-error without actual human efforts.

**Building Trust in Decision-making:** Trustworthiness is a cornerstone for the widespread acceptance of any system. Stakeholders and users seek control, transparency, and interactability. LLMs, adept at facilitating human-like interactions and explanations, enhance perceived trust with their inherently human-comprehensible interfaces that allow seamless and natural interaction with users.

Using the scenario of rangers protecting rhinos from poachers, intelligently using LLM can efficiently craft game-theoretic objectives, drawing from domain insights such as rhino movement patterns and past poaching incidents. It narrows the vast search space of the park using seasoned rangers' insights. As the ever-changing park landscape presents challenges, such as new poaching tactics or rhino migrations, the LLM dynamically refines strategies, learning and adapting with minimal manual oversight. Crucially, by articulating decisions in accessible terms, LLMs build trust with rangers, ensuring the devised strategies are both trusted and implemented.

To this end, we develop a comprehensive framework that harnesses the capabilities of LLMs to craft game-theoretic objectives tailored for multi-agent planning. Our approach centers around Generalized Nash Equilibrium Problems (GNEPs), a versatile game formulation that encompasses various conventional games, including dynamic games and normal form games. Bridging the gap between the LLM's natural language reasoning and structured game constructs like optimization problems or payoff matrices, we introduce a cost translation module. This module educates the LLM on interpreting agent behaviors through the lens of game theory, guiding it to suggest game formulations

that align closely with specified objectives. Once the LLM proposes a game, it is then processed by game solvers, such as nonlinear programming solvers, to determine the agents' actions. To complete the loop, we simulate the game's outcomes, using the results as feedback signals for the LLM, continuously refining and optimizing the proposed game structures.

We evaluate our framework in three benchmarks: (i) multi-agent motion planning for autonomous driving (ii) heterogeneous cooperative/competitive strategic reasoning for protecting rhino from poachers (iii) ThreeDWorld Multi-Agent Transport (TDW-MAT). In our autonomous vehicle experiments with non-cooperative quadratic costs, we seamlessly convert high-level descriptions to cost parameters for multi-agent behaviors. Our solution adeptly tunes intertwined parameters efficiently based on human feedback. Furthermore, we demonstrate our method can perform superior strategic planning comparing to other baselines in heterogeneous cooperative/competitive games in both the Stop-The-Poacher task and the TDW-MAT Challenge. To summarize, we contribute to: (i) A comprehensive framework with LLM integration to automate game-theoretic objectives design for multi-agent planning (ii) extensive evaluation across diverse benchmarks to demonstrate the capability to convert high-level descriptions into cost parameters and superior strategic planning in homogeneous/heterogeneous cooperative/competitive game scenarios.

## 2 RELATED WORK

**Game-theoretic planning.** A game-theoretic setting allows to model both adversarial and cooperative agents. In fact, scenarios such as autonomous driving, where robots need to interact with other intelligent agents, are fundamentally game-theoretic (Fisac et al., 2018; Trautman & Krause, 2010), (Dreves & Gerdts, 2018). Formalizing such interactions as a game, robots can weigh the impact of their decisions on the actions of other agents (Sadigh et al., 2016), even in competitive settings such as drone or car racing (Spica et al., 2018; Wang et al., 2021). Most game-theoretic works do not assume availability of a communication network among the agents and model the interactions as a game formulation. In (Peters et al., 2021; Laine et al., 2021; Le Cleac'h et al., 2021) data-driven methods are presented to infer objective functions of other agents online in game theoretic planning without communication. On the other hand distributed optimization approaches assume the existence of a communication network and solve the optimization based on the received communicated predictions (Rey et al., 2018), (Ferranti et al., 2018). They devise a communication and planning protocol in which all the agents share their predicted planned trajectories. One drawback of the former approach is that in an uncertain and dynamic environment such as autonomous driving setting, solving a game with a fixed structure may not be robust with respect to adversarial agents' behavior. Moreover, when there are adversarial agents involved in the game, communication is usually disabled, and the objectives of adversarial agents are usually hard to obtain. In this work, we show how we may infer objectives for multi-agent planning using large language models, in which case the planning is more robust and intelligent compared to existing data-driven methods.

**Large Language Models for Planning.** Large language models (LLMs), such as ChatGPT (Stiennon et al., 2020) (Gao et al., 2023), GPT-4 (OpenAI, 2023) (Katz et al., 2023), are powerful models for natural language understanding and inference. Recently, LLMs have been widely used in various robotic tasks, such as design (Singh et al., 2023), (Stella et al., 2023), (Vemprala et al., 2023), navigation (Shah et al., 2023a) (Shah et al., 2023b) (Dorbala et al., 2023), and planning (Chen et al., 2023) (Huang et al., 2022). Traditional planning methods require lengthy mapping and exploration, which is a major challenge since it may take a considerably long time to find a planning trajectory, especially in unfamiliar environments. However, humans can quickly navigate and plan in unfamiliar environments, which motivates the employment of LLMs for planning (Shah et al., 2023a) due to their human-level capability in scene understanding. LLMs planning in embodied environments requires planning skills, as well as the logic of planning. The solutions change over time in response to the agent's own and other agents' decisions (Huang et al., 2022). This problem is usually referred as a game. However, one obvious problem in planning using LLMs is that they generally struggle to understand the agent interactions that play a critical role in the performance one can achieve on a specific task in a multi-agent environment. In this work, we address this challenge by combining LLMs with game-theoretic optimization. Instead of directly planning actions, the LLM designs objectives for a game describing the interplay between agents. As a result, we can ensure the plans obtained do not neglect the interactive nature of the problem while harnessing the reasonning capabilities of LLMs.

## 3 LLMS AS GAME OBJECTIVE DESIGNERS

The framework we proposed does is not restricted to specific types of game formulations. It is a general approach that relies on the assistance of LLMs for the design of game objective parameters, which can be cost function parameters or simply payoff matrix entries. The applications we exhibit where such an architecture is useful include tuning game structures to obtain desired human behaviors in autonomous driving scenarios, as well as multi-agent planning where the LLM successively evaluates the payoff of actions available to agents in environments where the action space depends on the state of the system. The proposed architecture is depicted in Figure 1.

Given a multi-agent simulation environment, we observe the state of the system (or start from known initial conditions). The translation of the state of the system, the structure of the objectives of agents in the scene (available actions in normal form games, parameterized functions for continuous costs) can be done via an engineered prompt interfacing with the simulation environment, but can also accommodate a human in the loop for additional information or expressing human preferences. The LLM tunes objective parameters specified in the prompt, thus completing the information required to completely specify the game setup at a current time. The game can be solved with a suitable solver, generating the actions for each agent to execute. The simulator is then run until a new game structure is to be designed, and the process is repeated.

### 3.1 GENERALIZED NASH EQUILIBRIUM PROBLEMS

We consider Generalized Nash Equilibrium Problems (GNEPs) involving $N$ players $i \in \{1, \ldots, N\}$ over a horizon of $H$ time steps. An agent $i$'s state at time step index $t$ is denoted $\mathbf{x}_t^i \in \mathbb{R}^{n^i}$ and control input $\mathbf{u}_t^i \in \mathbb{R}^{m^i}$, with dimensions of agent $i$'s state and control $n^i$ and $m^i$. Let $\mathbf{x}_t = [\mathbf{x}_t^{1,\top}, \ldots, \mathbf{x}_t^{N,\top}]^\top \in \mathbb{R}^n$ denote the joint state and $\mathbf{u}_t = [\mathbf{u}_t^{1,\top}, \ldots, \mathbf{u}_t^{N,\top}]^\top \in \mathbb{R}^m$ denote the joint control of all agents at time $t$, with joint dimensions $n = \sum_i n^i$ and $m = \sum_i m^i$. We define player $i$'s policy as $\pi^i = [\mathbf{u}_1^{i,\top}, \ldots, \mathbf{u}_{H-1}^{i,\top}]^\top \in \mathbb{R}^{\tilde{m}^i}$ where $\tilde{m}^i = m^i(H-1)$ denotes the dimension of the entire trajectory of agent $i$'s control inputs. The notation $\neg i$ indicates all agents except $i$, for instance $\pi^{\neg i}$ represents the vector of the agents' policies except that of $i$. Also, let $X = [\mathbf{x}_2^\top, \ldots, \mathbf{x}_H^\top]^\top \in \mathbb{R}^{\tilde{n}}$, with $\tilde{n} = n(H-1)$, denote the trajectory of joint state variables resulting from the application of the joint control inputs to the dynamical system defined by $f : \mathbb{R}^n \times \mathbb{R}^m \to \mathbb{R}^n$ such that,

$$\mathbf{x}_{t+1} = f(\mathbf{x}_t, \mathbf{u}_t) \tag{1}$$

Over the whole trajectory we can express the above kinodynamic constraints with $\tilde{n}$ equality constraints,

$$D(X, \pi^1, \ldots, \pi^N) = D(X, \pi) = 0 \in \mathbb{R}^{\tilde{n}} \tag{2}$$

The cost function of each player $i$ can be parameterized by $\theta_i$, which determines its structure. The cost function depends on the agent $i$'s policy $\pi^i$ as well as on the joint state trajectory $X$, which is common to all players, such that $\forall i \in \{1, \ldots, N\}$,

$$J^{\theta_i}(X, \pi^i) = c_H^{\theta_i}(\mathbf{x}_H) + \sum_{t=1}^{H-1} c_t^{\theta_i}(\mathbf{x}_t, \mathbf{u}_t^i) \tag{3}$$

Notice that as player $i$ minimizes $J^{\theta_i}$ with respect to $X$ and $\pi^i$, the selection of $X$ is constrained by the other players' strategies $\pi^{\neg i}$ and the dynamics of the joint system via (2). In addition, the strategy $\pi^i$ could be required to satisfy constraints that depend on the joint state trajectory $X$ as well as on the other players strategies $\pi^{\neg i}$. This can be expressed with a set of $g$ inequality constraints,

$$C(X, \pi) \leq 0 \in \mathbb{R}^g \tag{4}$$

where $C : \mathbb{R}^{\tilde{n}} \times \mathbb{R}^{\tilde{m}} \to \mathbb{R}^g$. The GNEP we form is the problem of minimizing (3) for all players $i \in \{1, \ldots, N\}$ with respect to (2) and (4). More specifically,

$$\min_{X, \pi^i} J^{\theta_i}(X, \pi^i) \qquad \forall i \in \{1, \ldots, N\}$$
$$\text{subject to } D(X, \pi) = 0 \tag{5}$$
$$C(X, \pi) \leq 0$$

The solution to such a dynamic game is a generalized Nash equilibrium, i.e. a policy $\pi$ such that, $\forall i \in \{1, \ldots, N\}$, $\pi^i$ is a solution to (5) with the other players' policies given by $\pi^{\neg i}$ that is also solved by (5) for all $\neg i$. As a consequence, at a Nash equilibrium point solution, no player can improve their strategy by unilaterally modifying their policy.

Many familiar types of games fall under special cases of the GNEP formulation. Notably, matrix games can be seen as a special case of GNEPs, with a unit horizon, two players, and a cost structure dependency that can be described by a payoff matrix $M$. We can, thus, similarly parameterize the payoff matrix in a normal form game. In this case, asking the LLM to design the game parameters can be seen as having it fill in the entries of the payoff matrix. Achieving both tuning of continuous cost functions, as well as tabular cost evaluation, requires a capability to reason about actions and reactions.

### 3.1.1 GAME TRANSLATOR

We address the mechanism involved with obtaining LLM outputs that are amenable to conversion into games that can be directly handled by solvers.

The human assistant tuner setup we devise relies on a two-stage prompt architecture consisting of a motion descriptor followed by a cost function tuner. The method is an extension of the proposed single agent setup proposed by Yu et al. (2023). First, the motion descriptor instructs the LLM to interpret and translate the desired high-level user input behavior into a pre-defined template natural language description of the agents' behavior that is clearer for the LLM to decipher. Next, the cost function tuner prompts the LLM to turn the motion description into tuned cost functions in code form, which is subsequently inserted as is into the game definition code. We do not suggest any values or orders of magnitude.

When using our architecture for decision making in environments with dynamically changing discrete action options, prompting is designed to require the LLM to output payoffs for each combination of actions available to the agents at a specific planning step. This occurs in one prompting step, after which we parse the output to construct a normal form game. In two player scenarios this comes down to parsing the output into a matrix which is then solved to output the next Nash optimal action to be taken.

### 3.1.2 SIMULATOR FEEDBACK

In the tuning setup we assume the human operator provides short explanations of the behavior of agents in the simulator to the LLM. New parameters are then designed via the pipeline to be run again. When running the framework for decision making, we design a perception module, i.e. state to text scripts that take information from the simulator, such as agent positions and observations, current actions and so on, and maintain a history of the behavior, updating a prompt template. This ensures that all information required to assess options for a next planning step is made available to the LLM at evaluation time.

## 4 EXPERIMENTS

### 4.1 COST TUNING WITH LLMS FOR MULTI-AGENT MOTION PLANNING

In this section, we exploit the zero-shot reasoning capabilities of LLMs to ease the design of desired behavior in multi-agent motion planning. The coupling of optimisation problems in game-theoretic decision making makes it difficult to tune the cost parameters of different agents involved in the scene to achieve a desired collective behaviour. Using LLMs as desired motion descriptors and then cost function translators can significantly improve this tuning procedure. We use autonomous driving examples to illustrate the ease with which LLMs can help tune parameters to achieved desired road scenarios. Details on dynamics, constraints, cost functions and solver are provided in section A.0.1 of the Appendix.

### 4.1.1 NO OVERTAKE

We set two vehicles on the left side of a two lane highway. We wish to design a solution in which overtaking is only allowed from the left. Vehicle Green (going at 0.9 m/s) is 0.5 m behind Vehicle Red (going at 0.6 m/s). We wish for both vehicles to maintain their driving speed. Vehicle Red has to be reluctant to switch lanes. We desire to design a scenario where vehicle Green is forced to slow down to avoid crashing into vehicle Red. Figure 2 depicts the trajectories of the agents obtained by solving the game designed by our Game LLM architecture.

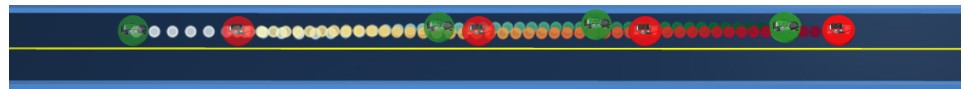

Figure 2: No overtake: Vehicle trajectories obtained by solving the dynamic game with the LLM generated cost parameters. Vehicle Green is in green and vehicle Red is in red. Marker size and color increase with time.

The LLM provides a set of parameters that achieve the desired behavior from the first attempt in zero-shot prompting fashion. Our architecture shows signs of multi-agent reasoning to tune these parameters from scratch, as it has no access to initial proposed values or orders of magnitude for the different parameters.

### 4.1.2 FORCE OVERTAKE

We set the two vehicles on a two lane highway. They are both on the driving on the left lane. Vehicle Green (going at 0.9 m/s) is 0.5 m behind Vehicle Red (going at 0.6 m/s). Both vehicles want to maintain their driving speed. Vehicle Red is reluctant to switch lanes. We prompt the LLM to design a scenario where vehicle Green overtakes vehicle Red via the right lane. Figure 3 depicts the trajectories of the agents obtained by solving the succession of games designed by our Game LLM architecture. After each execution, a human user gives a short description of the motion of the vehicles and asks the LLM to modify the parameters to get to the desired behavior.

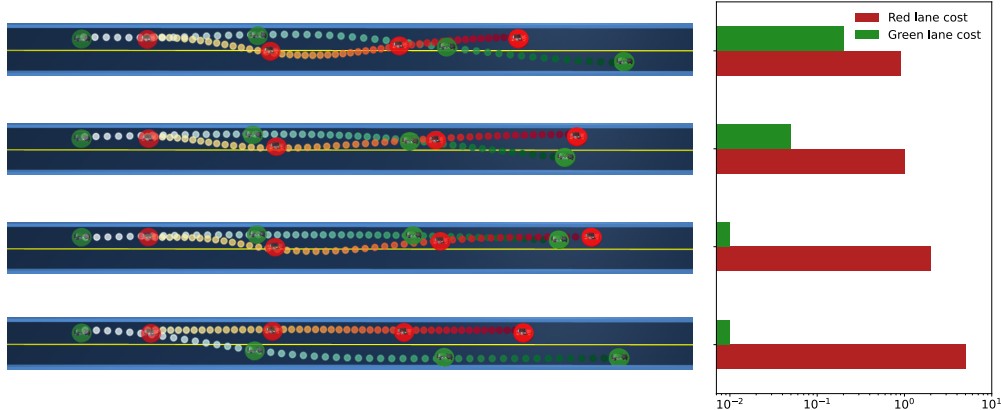

Figure 3: No overtake: Vehicle trajectories obtained by solving the dynamic game with the LLM generated cost parameters. Vehicle Green is in green and vehicle Red is in red. Marker size and color increase with time (vehicles driving from left to right). The trajectories are the iterations from earliest (top) to final (bottom) with user feedback to LLM as it tunes the objective parameters.

It takes our proposed architecture four attempts to get to the described scenario, which we argue is no more than what it would take a human to design such an experiment from scratch. Furthermore, we analyse the choice of parameters the LLM selects to tune and the value evolution with each iteration. In fact, the LLM pinpoints only two parameters that it modifies between runs; the cost parameters associated with deviations from the desired lane. Indeed, since the description suggests both agents wish to remain on the left lane and that overtakes should usually only occur from the

left, the desired lane for both agents is selected to be the left. The evolution of these parameters for both agents is depicted in the right bar plot in Figure 3.

An interesting observation is that in addition to both reducing the Green vehicles cost associated with staying on the left lane, and increasing that of the Red vehicle, it appears the tuner proceeds in exponential steps (the bar plot is in log scale, the lane cost for Green is successively 0.2, 0.05, 0.01, 0.01 whereas the lane cost for Red is 0.9, 1.0, 2.0, 5.0). Our LLM tuner thus exhibits an efficient tuning approach while showing correct understanding of human feedback and identifying parameters of significance. It is also notable that the tuner is capable of tuning multiple values simultaneously without resorting to a complete decoupling approach.

Our LLM-based architecture designed to reason about multi-agent behavior proves to be a useful assistant for the design of objectives in game-theoretic motion planning scenarios.

## 4.2 STOP-THE-POACHER

Stop-the-Poacher is a task that involves heterogeneous robot agents in a cooperative/competitive environment. The task is designed to require coordinated team planning of agents with different attributes to contain an adversary. The simulation is set up in TDW (Gan et al., 2021a), where a team consisting of a Responder 4x4 SUV and a Drone are tasked with protecting a Rhino from a Poacher (humanoid).

### 4.2.1 TASK DESIGN

The environment consists of an open square simulation area. The Poacher is given privileged information about the location of the Rhino and heads directly towards it. The locations of the Rhino and Poacher are unknown to the Responder and Drone who have to first coordinate the exploration of the area to locate other agents that enter their perception radius. The mission is considered a success if the Responder can secure the Rhino (by reaching it before the Poacher) or if it can intercept the Poacher before it gets to the Rhino. The Drone cannot physically intercept or secure, but can only gather information. The Rhino and Poacher are initially spawned at random positions on the map. The Drone and Responder both start at the center of the map in each simulation run. They both move at the same speed and are faster than the Poacher. We assume the Responder and Drone share information about their positions and the positions of agents they perceive.

We calibrate the map size and agent moving speeds for the success rates to be reasonable using a baseline heuristic. We segment the map into 9 equal areas in a 3x3 square grid pattern. We reference the different areas by their cardinal direction with respect to the central region, which we denote capital C (for center). For example, the region to the east of C is denoted E.

Although the decision agents do not share the same physical capabilities, they share the same action space in our setup. Both the Responder and the Drone can explore one of the 9 regions available. Upon detection of the Poacher or the Rhino, they can also move towards either.

The LLM is tasked with evaluating the payoff value of each pair of actions at each given time, and the pair maximising the team's chances of success is executed. A new decision is triggered in two scenarios: if an area has been completely searched or if a new agent is perceived.

### 4.2.2 PERFORMANCE METRIC

The metric by which we evaluate a simulation run is the mission success rate. Indeed, all runs that end with an unintercepted Poacher reaching the Rhino before the Responder are mission failures. In the opposite case, if the Rhino is secured or if the Poacher is caught we consider the mission to be a success. Policies are evaluated over a sample of initial simulation conditions.

### 4.2.3 RESULTS

Four different planners are evaluated on the Stop-the-Poacher task. We first let the agents take a uniformly randomized action from the list of available actions at the current state. We also develop a heuristic that reasons as follows: while neither of the Poacher or the Rhino are spotted, the agents explore opposite directions of the area (in order: S, SW, W, NW for the Responder, and N, NE, E,

SE for the Drone). When one of the Rhino or Poacher are spotted, both the Drone and Responder begin to make their way towards it. If the second agent is perceived, the response team make their way towards the closest of the two.

We also propose a Direct LLM decision planner. With knowledge of the positions of the Drone and Responder, past actions and explored areas, and presence or not of the Rhino and Poacher in the perception field, the LLM is asked to select a pair of admissible actions in a zero-shot manner, without any reasoning artifacts or additional design.

We compare these baselines to our Game LLM architecture, which prompts the LLM to output a score for each pair of actions available to the agents at a given state. The scores can then be combined to constitute a matrix, the argmax of which is the optimal action as evaluated by the LLM in our cooperative setting.

Both LLM architectures are instantiated with GPT-4. We access GPT-4 from the OpenAI Python API and with temperature equal to 0.0, top-p set to 1, and a max number of tokens of 1024.

The results, in terms of number of runs ending in a poach, in an interception of the Poacher, or in a secured Rhino as well as the total success rates achieved by the different planners are presented in Table 1.

Table 1: Results for the Stop The Poacher task. (N=64).

| ALGORITHM | POACHES | INTERCEPTIONS | SECURED | SUCCESS RATE |
|---|---|---|---|---|
| Random action | 55 | 3 | 6 | 14 % |
| Heuristic | 12 | 33 | 19 | 81.25 % |
| Direct LLM | 20 | 26 | 18 | 68.7 % |
| Game LLM | 9 | 30 | 25 | 85.94 % |

The heuristic planner that serves as a principled baseline achieves a success rate of just over 80%. We present a random action planner to exhibit the amount of reasoning required to achieve the task in a successful way. It completes the task on about 14% of runs. The Direct LLM architecture can achieve the task just under 70% of the time. Thus, it does exhibit some reasoning capability to decide simultaneous actions for coupled agents. However, its performance is below that of a simple hand designed human heuristic. Our proposed architecture improves the multi-agent planning performance of the LLM, completing the task for close to 86% of runs, a higher return than that of the baseline heuristic. By design, the Game LLM formulation is better adapted to this type of problem, as it encourages the LLM to reason about outcomes of combinations of actions instead of solely predicting a next decision.

## 4.3 MULTI-AGENT TRANSPORT CHALLENGE

We test our framework on the ThreeDWorld Transport Challenge (Gan et al., 2021b), more specifically on a multi-agent extension with additional objects and containers, and more realistic objects placements. The experiment is named ThreeDWorld Multi-Agent Transport (TDW-MAT) (Zhang et al., 2023). The simulation is also built on top of the general-purpose virtual world simulation platform TDW (Gan et al., 2021a).

### 4.3.1 TASK DESIGN

The agents are tasked with transporting the largest number of target objects possible to goal positions, using containers as tools (otherwise agents can transport only two objects at a time). The actions available to agents include exploring a room, moving to an other room, manipulating objects (pick up, drop), and using containers. The task is simplified from the setup presented in Zhang et al. (2023). Indeed, we assume centralized information and decision making, and thus remove the need for communication actions.

We select 6 scenes from the TDW-House dataset and run 2 samples of each of the 2 types of tasks in each of the scenes, thus building a test set of 24 episodes. Every scene has 6 to 8 rooms, 10 objects,

and 4 containers. An episode is terminated if all the target objects have been transported to the goal position or the maximum number of frames (3000) is reached.

### 4.3.2 PERFORMANCE METRICS

We first define the Transport Rate (TR) as the fraction of the target objects successfully transported to the goal position. This will serve as our metric for the task. We can also reinterpret the metric in terms of Efficiency Improvement (EI), that being with respect to a single agent attempting the task, which is computed according to $\text{EI} = \frac{1}{N}\sum_{l=1}^{N}\left(\text{TR}_{multi,i} - \text{TR}_{single,i}\right)/\text{TR}_{multi,i}$, with $\text{TR}_{single,i}$ denoting the single agent's transport rate for episode $i$, and $\text{TR}_{multi,i}$ denoting the two-agent transport rate for episode $i$.

### 4.3.3 RESULTS

We test our Game LLM setup against a Direct LLM planner that has to pick the next pair of actions directly from the list of available actions to each player. Designing a heuristic for this setup is far from straighforward so we only compare our approach to a non-game reasoning LLM. The results are presented in Table 2.

Table 2: Transport Rates and Efficiency Improvements of the Direct LLM method vs our Game LLM on the TDW-MAT task.

| ALGORITHM | TRANSPORT RATE | EI |
|---|---|---|
| Direct LLM | 0.64 | 19% |
| Game LLM | 0.67 | 22% |

The Direct LLM approach achieves a Transport Rate of 0.64 and is outperformed by our Game LLM planner (0.67). We believe the expected improvement with respect to the direct method is mitigated by the migration of the task from a decentralized setup (may need additional adaptation to truly capture the mecanisms we seek to act upon). However, this remains in line with the remarks made above about the more explicit reasoning about interaction that our architecture imposes on the LLM.

## 5 DISCUSSION

In this work, we propose a general framework to automate the design of game objectives for multi-agent planning that capitalizes on Large Language Models' reasoning capabilities without compromising on the interactive dimension of the problem.

Our methodology shows promise as a human assistant for the design of scenarios with desired interactive behavior. We experiment with designing autonomous vehicle scenarios with non-cooperative quadratic cost structures and show the ease with which high-level descriptions can be converted into cost parameters that achieve the sought multi-agent conduct. We also notice that through simple human exchange, our solution proves to be an efficient tuning aide, understanding brief feedback and efficiently adjusting coupled parameters.

The proposed architecture also proves effective as a planner in dynamical multi-agent environments where players are faced with a succession of decisions. Evaluating it in two such simulation settings, our proposed planner surpasses both the hand designed heuristic and the direct action selection approach on the Stop-The-Poacher task, as well as exceeding the latter approach in the TDW-MAT Challenge.

One major limitation of our approach is scalability. Although one can instruct the LLM to reduce its search space by only considering a subset of actions that are the most sensible for each agent before evaluating combinations, it is clear that for large systems the number of action combinations grows exponentially with the number of agents. However we argue that in most robotics scenarios, it is seldom the case that more than a handful of agents interact at once.

Extensions to such scenarios beyond two agents and competitive settings with partial observations of the decision adversary are directions of future work.

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

# A APPENDIX

### A.0.1 DETAILS: AUTONOMOUS DRIVING EXAMPLE

The dynamic game solver we use is ALGAMES (Cleac'h et al., 2019), which handles trajectory optimization problems with multiple actors and general nonlinear state and input constraints. The vehicles obey nonlinear unicycle dynamics. The state of a vehicle $\mathbf{x}_t^i$ comprises of its 2D position (*px* horizontal, *py* vertical), its heading angle and scalar velocity. The control input $\mathbf{u}_t^i$ comprises of the angular velocity and scalar acceleration.

The dynamics constraints at time *t* consist in following the system dynamics given by equation 1, with *f* being unicycle dynamics in our driving simulations. We also enforce collision-avoidance constraints on the trajectories, by modelling collision zones of the vehicles by circles or radius *r*, such that, at any time step *t*, $\|\mathbf{p}_t^i - \mathbf{p}_t^j\|_2^2 \geq r^2, \quad \forall i,j \in \{1,\ldots,N\}$. In addition, we require the vehicles to remain on the road, by constraining the distance between each vehicle and the closest point $\mathbf{q}$ on each boundary *b* to remain larger than the collision radius *r*, $\|\mathbf{p}_t^i - \mathbf{q}_b\|_2^2 \geq r^2, \quad \forall b, \forall i \in \{1,\ldots,N\}$ where $\mathbf{p}_t^i = [px_t^i, py_t^i]$ contains the plane coordinates of the agent *i* at time *t* extracted from the complete state vector $\mathbf{x}_t^i$. Thus, the autonomous driving problem is formalized via non-convex and non-linear coupled constraints.

The cost structure considered is quadratic, penalizing the distance to the desired final state $\mathbf{x}_f$ and the use of controls,

$$J^{\theta_i}(X, \pi^i) = \sum_{t-1}^{H-1} \frac{1}{2}(\mathbf{x}_t - \mathbf{x}_f^{\theta_i})^\top \mathbf{Q}^{\theta_i}(\mathbf{x}_t - \mathbf{x}_f^{\theta_i}) + \frac{1}{2}\mathbf{u}_t^\top \mathbf{R}^{\theta_i}\mathbf{u}_t + \frac{1}{2}(\mathbf{x}_H - \mathbf{x}_f^{\theta_i})^\top \mathbf{Q}_f^{\theta_i}(\mathbf{x}_H - \mathbf{x}_f^{\theta_i}), \quad (6)$$

where $\mathbf{Q}^{\theta_i}$, $\mathbf{R}^{\theta_i}$ and $\mathbf{Q}_f^{\theta_i}$ represent state, input and final state penalization weight matrices, respectively. They can all be parameterized such that the LLM decides what values to encode in them. The same applies to the final desired state $\mathbf{x}_f^{\theta_i}$. This cost function depends only on the decision variables of vehicle *i*, as players' behaviors are only coupled through collision constraints. Thus, although knowledge of other agents' intentions does not intervene in the individual cost agent *i* is optimizing for, it does however determine the trajectories of others, and subsequently the collision constraints agent *i* has to satisfy.

The parameters we ask the LLM (GPT-4) to select are the target driving speed, the target lane to drive in (both in $\mathbf{x}_f^{\theta_i}$), the cost associated with deviations from the desired speed, the cost associated with deviations from the desired lane, and the cost associated with deviations from the desired driving angle, which is set to be parallel to the road (part of $\mathbf{Q}^{\theta_i}$).

