# OpenReview forum: "Large Language Models Can Design Game-Theoretic Objectives for Multi-Agent Planning"
_ICLR.cc/2024/Conference — ICLR 2024 Conference Withdrawn Submission_

### Official Review · Reviewer_qW2i · 2023-10-27

**Soundness:** 2 fair
**Presentation:** 2 fair
**Contribution:** 3 good
**Rating:** 3
**Confidence:** 4

**Summary:**

This paper proposes a framework that exploits the power of large language models to tailor game-theoretic objectives for multi-agent planning. This framework focuses on the generalized Nash equilibrium problem (GNEP). The entire framework is divided into two parts, namely game translator and game solver. First, the motion descriptor in the game translator converts the state into natural language input to the LLM, and then the cost function tuner prompts the LLM to turn the motion description into tuned cost functions in code form. The game solver will give the solution to the corresponding game and let the agent execute these policies in the simulator. The final results will be fed back to LLM to redesign the game. The proposed framework in this paper was experimented on three tasks,  including Autonomous Driving, Stop-The-Poacher and Multi-Agent Transport.

**Strengths:**

1. The writing is easy to understand.
2. The proposed Game LLM framework was experimented in multiple scenarios.
3. The visualization of experimental results is done well, and the results are easy to follow.
4. It is relatively novel to design the objective function in the game through a large language model.

**Weaknesses:**

1. The implementation details of the framework and the details of the experimental process are not sufficiently described. The experimental results feel difficult to reproduce.
2. The experimental results are unconvincing. For example, in the ThreeDWorld Transport Challenge, the performance of Game LLM is very close to that of Direct LLM.
3. I do not see the role of the Simulator Feedback process in the Stop-The-Poacher and Multi-Agent Transport scenarios.

**Questions:**

1. What are the specific prompts input to LLM in each stage? There is almost no mention of prompt-related information in the entire text.
2. What is the specific role of Simulator Feedback in the Stop-The-Poacher and Multi-Agent Transport scenarios? And what exactly is the game solver in these two scenarios?
3. In the ThreeDWorld Transport Challenge, how to ensure that the performance difference between the two methods is not caused by different prompts, or not caused by the randomness of the large language model (even if temperature is 0)?
4. It seems that the description of GNEPs in Section 3.1 is redundant, because there is actually not much game-related settings involved.

---

### Official Review · Reviewer_CZtQ · 2023-10-30

**Soundness:** 3 good
**Presentation:** 3 good
**Contribution:** 2 fair
**Rating:** 5
**Confidence:** 3

**Summary:**

This paper proposes a framework that uses LLMs to control game parameters in order to generate human-desired behaviours of agents. Specifically, an LLM is used to pick either cost-function parameters of payoff matrix entries for N player games.

**Strengths:**

- Overall, I think the paper is well-presented and the proposed framework is generally easy to understand. In addition, the idea proposed is novel and does make sense for a variety of contexts.
- The experiments are reasonably well-designed and are effective in demonstrating some of the basic capabilities of what the framework can do.

**Weaknesses:**

In my opinion there are two more problems with the framework that both arise in more complex environments:

1) In the environments tested, certain types of desired behaviour of the agents are fairly obvious and providing some specific details is simple (e.g. only overtake in X lane, do not overtake etc...). However, I envisage this becoming a potential problem in larger / more complex domains. For example, whilst some form of desired behaviour will always be fairly easy to specify - the increasing complexity of how one would reach that desired behaviour may require more specific details provided by the human. Then the bottleneck instead becomes the human running the experiment instead of the LLM tuner.

2) The authors make note of this in their discussion, however I think it is important that they spend more time discussing the scalability problem of this framework. In particular, what are some techniques / methods that could be used in order to reduce the exponentially growing number of action combinations when either number of players increases or the number of possible actions for each player increases?

**Questions:**

I would be happy if the authors could address my points made in the weaknesses section. Primarily:

1) The bottleneck of defining desired behaviours from the human side in games that are complex

2) Some more discussion on the problem of scalability, and some suggested approaches to how the problem could be resolved.

---

### Official Review · Reviewer_ZJKY · 2023-10-31

**Soundness:** 2 fair
**Presentation:** 2 fair
**Contribution:** 2 fair
**Rating:** 3
**Confidence:** 4

**Summary:**

This paper leverages LLMs to design the game objectives, thus improving the performance of different methods. Experiments on multi-agent motion planning, protecting rhino, and multi-agent transport demonstrate the effectiveness of the proposed methods.

**Strengths:**

I generally agree that incorporating LLM in game theory is a general framework for improve the game-theoretic methods. The problem considered in this work is interesting and the methods are promising.

**Weaknesses:**

This paper is hard to follow, the technical details are not fully elaborated. It is hard to evaluate the technical contributions of this paper and whether the proposed methods are general enough as claimed. (More in the questions section)

**Questions:**

There are several main questions I want the authors to address in the next version of the paper.

1. The flow of the paper is hard to follow. Suggestions include:
a. A preliminary section is needed after section 2. Actually section 3.1 should be the preliminary section. section 3.1.1 and section 3.1.2 should be subsection, rather than subsubsection. Please add the preliminary section to define the problem, the objective, the considered solution concept for this work, which is very helpful for reviewers and readers to understand the paper.
b. For the introduction section, you introduce four necessities about introducing LLM into game-theoretic approaches, e.g.,
Automated Game Design and Harnessing Domain Expertise, however, these points are not relevant to the methods you proposed and are not shown in the experiment section. So please focus on your problem, and do not discuss so broad topics. Good experiments are better than words to demonstrate the necessities of the methods.
c. The technical details are not clear. There are only three parts introducing the techniques, i.e., 'To this end' para in section 1, and sections 3.1.1 and 3.1.2. Most of them are not technical words, I cannot find the technical descriptions of your methods. Even not equations, please provide the high-level ideas of the methods.



2. About techniques (based on my understanding, as the paper is not fully elaborated).
a. When you ask the LLM to design the parameters, what are the prompt? what is the objective when you design these prompts? does this design methods generalizable across different multi-agent scenarios?
b. How to handle the limited context length of LLM? Say, when you consider a very complicated, that may exceed the maximum length of the LLM's input, how to handle this? Or, what is the max current length of the prompts across different scenarios in this paper?
c. I cannot fully understand what are the motion descriptor and the cost function tuner? please add more details about the two modules, including illustrative figures and examples.
d. For the simulator feedback, how to include these feedback into the prompt for the iterative design? is it similar to COT or Reflexion?  Please give more details. Besides, normally when given the parameters of the game, we can receive the final performance, what is the difference between your simulator feedback comparing with the traditional setting? If they are same, why the simulator feedback still need to be highlighted in the paper?

3. About the experiments.
a. why the three scenarios are chosen? Honestly, the three scenarios are not widely considered in the game-theoretic literature. Why not choosing poker or board game?
b. The baselines are random actions, heuristic, direct LLM. There would be some domain-specific solvers about the problem, which can be viewed as the upper bound of the LLM's performance. Maybe please add them into the baselines, so we can evaluate the proposed methods against the sota methods.

---

### Official Review · Reviewer_ooPj · 2023-11-06

**Soundness:** 2 fair
**Presentation:** 2 fair
**Contribution:** 2 fair
**Rating:** 3
**Confidence:** 4

**Summary:**

This article proposes a system where a language model can be used to produce an interpretation of a complex system in a way that is amenable to game theoretic analysis, and thus, strategic optimisation. The authors show that such a system can be much more efficient at solving complex problems and require a lot less human intervention that traditional human feedback systems. The authors show haw their system performs well across 3 decision making multi-agent environments: overtaking in traffic, stop the poacher, and multi-agent transport. The game theoretic component is conceptualised as a generalised Nash equilibrium, which is a formulation that admits normal and extensive form games among others.

**Strengths:**

The article is attempting to leverage the benefits of strategic thinking derived from game theory with the intuition and general knowledge of language models. The domain problems and methodology are interesting and merit study. The idea of using language models to understand complex situations is a growing field of study, and this work extends it in multi-agent situations.

**Weaknesses:**

This might be a good article, but there are a number of aspects that make it very hard to evaluate how significant the contributions are.

My main concern is that the action space of the language model is so small as to be effectively a trivial search problem. I wouldn't be surprised if full ennumeration of the possible strategic space would cost less than a single inference on the language model. Even if not concerned about computational cost (one could argue that it is still worth it knowing that LLMs are capable to reasoning about strategic thinking), the search spaces appear to be so small that it is unclear that the LLM is actually reasoning to any significant degree. I worry that most of the apparent reasoning is built into the agents, which end up being controlled by the LLM goal setting.

Another aspect that requires elaboration is the prompt template system. This template is tuned in some way over iterations that mostly are due to human feedback. However, we do not get to see prompt, so it is quite possible that everything of interest went into a very cleverly engineered prompt that causes the appearance of strategic behaviour to emerge.

A final critical detail that appears absent is how are the actual agents in the environment trained. They must take the outputs from the LLM as strategic guide, but this requires a mapping of an abstract (and potentially high-level) action, into a policy to be deployed. Perhaps the authors train a policy that can be conditioned on the possible goals chosen by the LLM. If so (and it seems from some of the descriptions that this is the case), then the generality of this algorithm is very limited: it requires setting special ways of extracting text from environments, and also requires taking high-level goals and implementing them as low-level policies that can be easily conditioned on those goals. The fact that this can be done with an LLM in a very low-dimensional space might be rather trivial if so. This seems the case as the heuristic results appear very close to the LLM ones.

The system appears to operate in two phases, although this is never explicitly described: Tuning: which involves using human explanations of behaviour of lower level agents and that is somehow resulting in a parametric prompt; and Decision making: where the system is behaving autonomously, receiving text observations from the environment that it can use to issue commands to the lower level agents.

**Questions:**

The authors should define what they mean by $D$ in equation (2) and $c$ in equation (3). Those are never defined.

The explanations of Section 3.1.1 and 3.1.2 are not provided in sufficient detail or clear enough.

I would also like to know how well this approach scales to more strategic options. What happens when the LLM must output a 10x10 strategy tensor to then solve with game theory? I suspect an LLM would have a _very_ hard time getting such an incentive structure right.

The authors should provide more details about the inputs and outputs of their model.